# Improved Performance of Solid Polymer Electrolyte for Lithium-Metal Batteries via Hot Press Rolling

**DOI:** 10.3390/polym14030363

**Published:** 2022-01-18

**Authors:** Poonam Yadav, Seyed Hamidreza Beheshti, Anish Raj Kathribail, Pavlo Ivanchenko, Joeri Van Mierlo, Maitane Berecibar

**Affiliations:** MOBI Research Group, Department of Electric Engineering and Energy Technology (ETEC), Vrije Universiteit Brussel, Pleinlaan 2, 1050 Brussels, Belgium; Seyed.Hamidreza.Beheshti@vub.be (S.H.B.); Anish.Raj.Kathribail@vub.be (A.R.K.); Pavlo.Ivanchenko@vub.be (P.I.); Joeri.Van.Mierlo@vub.be (J.V.M.); Maitane.Berecibar@vub.be (M.B.)

**Keywords:** solid polymer electrolyte, hot press rolling, ionic conductivity, PVDF-HFP, crystallinity of polymer, grain boundary, lithium-metal battery

## Abstract

Solid-state batteries (SSBs) are gaining attention as they promise to provide better safety and a higher energy density than conventional liquid electrolyte batteries. Solid polymer electrolytes (SPEs) are promising candidates due to their flexibility providing better interfacial contact between electrodes and the electrolyte. However, SPEs exhibit very low ionic conductivity at ambient temperatures, which prevents their practical use in batteries. Herein, a simple and effective technique of hot press rolling is demonstrated to improve ionic conductivity and, hence, the performance of polyvinylidene fluoride-co-hexafluoropropylene (PVDF-HFP)-based solid polymer electrolyte. Applying hot press rolling to the electrolyte membrane induced structural changes in the grain boundaries, which resulted in a reduction in the crystallinity of the material and, hence, an increase in the amorphous phase of the material, which eased the movement of the lithium ions within the material. This technique also improved the surface of the membrane, making it homogeneous and smoother, which resulted in better interfacial contact between the electrodes and electrolyte. Electrochemical tests were carried out on electrolyte membranes treated with and without hot press rolling to evaluate the effect of the treatment. The hot pressed electrolyte membrane showed significant improvements in its ionic conductivity and transference number. The cycling performance of the LFP/Li batteries using a hot press rolled electrolyte was also evaluated, which gave a specific discharge capacity of 134 mAh/g at 0.1 C. These results demonstrate that hot press rolling can have a significant effect on the electrochemical performance of solid polymer electrolytes.

## 1. Introduction

Climate change and global warming are real issues that demand immediate attention. The transportation sector, which operates on fossil fuels, contributes 16.2 percent of carbon emissions [1]. Replacing fossil fuels with electricity in transport vehicles is a promising way to reduce carbon emissions. The current battery systems cannot fulfill the demands of electric vehicles in terms of energy density, power density, and safety. In addition, conventional batteries contain an organic liquid electrolyte which is flammable and causes serious safety issues. Solid-state batteries (SSBs) are a promising solution as they result in fewer safety concerns compared to conventional batteries with liquid electrolytes as solid electrolytes are thermally more stable [2]. The use of solid electrolytes also paves the way for the use of lithium metal as the anode, which is not possible with liquid electrolytes because of their high reactivity with lithium metal [3]. Lithium metal is a favorable anode material owing to its high energy density (3680 mAh/g), which is 10 times better than the specific capacity provided by conventional graphite anodes [3,4]. Irrespective of all these advantages, several challenges such as the low ionic conductivity of solid electrolytes, high interfacial resistance and instability between electrodes and solid electrolytes remain to be addressed before the use of SSBs becomes practical [5,6].

Generally, solid electrolytes can be classified into three categories: inorganic solid electrolytes (ISEs), solid polymer electrolytes (SPEs), and composite solid polymer electrolytes (CSPEs) [7]. Each solid electrolyte has its own advantages and challenges to overcome. Inorganic solid electrolytes (sulfides, oxides, phosphates) usually have high ionic conductivity but face the challenge of high solid–solid interfacial resistance between the electrodes and electrolyte [8]. The application of pressure at the cellular level is required for batteries using ceramic solid electrolytes. Solid polymer electrolytes (SPEs) consist of a polymer matrix and lithium salts, and possess excellent flexibility, good solid–solid interfacial contact, and easy processability [8,9], but present low ionic conductivity (<$10^{-4}$ S/cm) in ambient conditions, as well as having inferior thermal and electrochemical stability [7]. Composite solid polymer electrolytes (CSPEs) are made by adding inorganic ceramic as filler into solid polymer electrolytes [10], and they promise to inherit the advantages of both SPEs and inorganic fillers, such as better ionic conductivity, flexibility, and mechanical strength.

In solid polymer electrolytes (SPEs), PEO is the most extensively studied polymer matrix, mainly due to its ability to dissolve alkali-metal salts, low glass transition temperature, and non-toxicity [11]. On the other hand, PVDF-HFP is a good candidate for a polymer matrix due to its low crystallinity, high dielectric constant (ε = 8.4), which helps to dissolve lithium salts to a greater extent, and good mechanical stability over a wide range of temperatures [12,13,14,15]. However, ionic conductivity is the most important indicator when evaluating the performance of an electrolyte, which is determined by the mobility of the charge carriers. The ionic conductivity of solid polymer electrolytes is severely affected by the large crystalline domains present in the polymer chains [16]. Crystallinity is a fundamental characteristic of polymeric systems which defines mechanical, thermal, optical, electronic, and transport properties [17,18,19]. A high crystallinity limits the movement of lithium ions [20,21] and, hence, enhances the ionic conductivity of SPEs. Several approaches, such as adding plasticizer and incorporating ceramic filler, have been widely explored [22,23,24,25,26] and demonstrated to successfully disrupt this crystallinity. However, a simple processing technique can also greatly influence the quality of the electrolyte. For example, Wang et al. [27] used an ultrasonic vibrational technique to suppress the crystallinity of solid polymer electrolytes, which resulted in improvements in the ionic conductivity of the electrolytes.

In this work, a solid polymer electrolyte made of a Polyvinylidene fluoride-co-hexafluoropropylene (PVDF-HFP) polymer matrix and a Bis(trifluoromethylsulfonyl)amine (LiTFSI) lithium salt was studied. A simple and effective technique of hot press rolling was implemented to reduce the crystalline matrix present in the polymer chains and to improve the performance of the SPE. In solid polymer electrolytes, lithium ions are thought to be transported through grain boundaries and an amorphous phase [28]. The use of hot press rolling with maintained pressure and temperature on the solid electrolyte membrane can break grains and reform grain boundaries, which increases the amorphous region; hence, it has the potential to be applied to SPEs to reduce the crystallinity of the polymers and improve the ionic conductivity of the electrolytes. Furthermore, detailed electrochemical analyses of the hot press rolled and non-hot press rolled SPEs were carried out to evaluate their performances in lithium-metal battery application.

## 2. Materials and Methods

### 2.1. Materials Used

Polyvinylidene fluoride-co-hexafluoropropylene (PVDF-HFP) (average relative molecular mass $M_{w}$ of ~455,000), Bis(trifluoromethylsulfonyl)amine (LiTFSI) lithium salt (99.95%), Lithium iron phosphate (LiFePO_4_), and N-Methyl-2-pyrrolidone (NMP) (99.95%) were purchased from Sigma-Aldrich and used as received. Carbon (Super-p) was purchased from Alfa Aesar. The chemical structures of the polymer and lithium salt used in the preparation of the solid polymer electrolytes is shown in Figure 1.

### 2.2. Methods

#### 2.2.1. Preparation of Solid Polymer Electrolyte Membranes

The solid polymer electrolyte membranes were prepared by a solution casting technique. The polymer matrix (PVDF-HFP) to lithium salt (LiTFSI) ratio was kept constant at 2:1 wt%. LiTFSI (1.25 g) was dissolved in NMP (10 mL) and stirred on a heating plate at 50 °C for 2 h (Figure 2a,b). Thereafter, PVDF-HFP (2.5 g) was added and stirring continued for 12 h (Figure 2c). The resultant slurry was poured into a petri dish, and the solvent was allowed to evaporate slowly at room temperature in a dry room (dew point: −50 °C ) for 12 h (Figure 2d,e). Residual solvents were further removed by drying in a vacuum oven at 80 °C for 16 h (Figure 2f). As a result, the solid polymer electrolyte (SPE) self-standing films were obtained (Figure 2g). Similarly, the desired amount of SPE was fabricated and hot press rolling treatment was performed under controlled conditions. From here onwards, the membranes without hot press rolling treatment are named SPE and those with hot press rolling treatment are named R-SPE. The final thicknesses of both types of electrolyte membrane were kept the same, within the range of 85–100 μm. Membranes were then cut into circular disks with diameters of 18 mm and were stored in a glovebox for further testing. The final images of a membrane are shown in Figure 3.

#### 2.2.2. Hot Press Rolling Treatment of Solid Polymer Electrolytes

The equipment (Figure 4) used to perform the hot press rolling treatment of the as-prepared solid polymer electrolyte membranes was a Hot Rolling Press (MSK-HRP-MR100DC, MTI). The membranes were treated under controlled conditions (Temperature = 60 °C, rolling speed = 4 rpm (speed display units in instrument), rolling gap = 0.1 mm). This treatment of the membranes is thought to deform the grain boundaries of the polymer structure, as shown in Figure 5, and more grain boundaries are created.

#### 2.2.3. Catholyte Preparation

Catholyte was prepared through a conventional doctor-blade method. A homogeneous slurry was prepared with the lithium iron phosphate (LFP), carbon (Super-P), and solid polymer electrolyte (SPE) solution in a mass ratio of 80:10:10. A homogeneous slurry was prepared and then coated on aluminum foil with the help of the doctor blade, with a wet slurry thickness of 200 µm. The electrode sheets were dried at room temperature for 2 days and then cut into 15 mm diameter disks. All preparations and electrode cutting were performed in a dry room with an approximately −50 °C dew point. The disks were finally dried at 100 °C in vacuum for 16 h. The active material (LFP) mass loading of the as-prepared electrodes was ~3.5 mg/cm^2^.

### 2.3. Characterization Methods

#### 2.3.1. Electrochemical Impedance Spectroscopy (EIS)

Electrochemical impedance spectroscopy (EIS) measurements were implemented on a VSP Biologic in a frequency range of 10 kHz–10 Hz to measure the ionic conductivity of the hot press rolled and non-hot press rolled solid polymer electrolytes.

#### 2.3.2. Direct Current (DC) Polarization

The lithium ion transference numbers of the SPE and Cal-SPE were measured by performing AC impedance and DC polarization of a Li/SPE/Li symmetrical cell in a coin cell (CR2032) setup. EIS measurements were performed before and after the DC polarization test. The voltage applied for DC polarization was 10 mV.

#### 2.3.3. Linear Sweep Voltammetry (LSV)

The electrochemical window was determined by linear sweep voltammetry (LSV) using a Li/SPE/SS configuration in a coin cell (CR2032). The tests were carried out at a scan rate of 0.1 mV/s over a potential range of 0–5.5 V.

#### 2.3.4. Symmetrical Cell Cycling

To see the stability of the electrolyte membranes against Li dendrite propagation, galvano-static cycling tests were carried out on symmetric cells (Li/SPE/Li), in a coin cell (CR2032), at constant current densities of 0.1 mA/cm^2^, 0.2 mA/cm^2^, and 0.5 mA/cm^2^ at 60 °C.

#### 2.3.5. Galvano-Static Charge–Discharge Cycling

The electrochemical performance of the solid polymer electrolytes was tested with LFP/Li half cells with as-prepared LFP catholyte and Li metal as the anode. Coin cells (CR2032) were assembled in an argon-filled glovebox (BRAUN, Germany, H_2_O and O_2_ ≤ 1 ppm). Charge and discharge tests were carried out over a 2.8–3.8 V voltage range at 60 °C. Two formation cycles at C/50 were carried out before testing at high C-rates.

## 3. Results and Discussion

### 3.1. Ionic Conductivity

To find the ionic conductivity of the solid polymer electrolytes SPE and R-SPE, cells with the configuration Li/electrolyte/Li in a coin cell setup were fabricated, and EIS tests were conducted at different temperatures of 10 °C, 20 °C, 30 °C, 40 °C, 50 °C, and 60 °C (Figure 6).

The bulk resistance values were obtained by fitting the equivalent circuit (Figure 7) using Z-fit in EC-Lab software. The fitting of the experimental data to this circuit model allows the estimation of bulk resistances (R2). The results of this fitting demonstrate that the R-SPE electrolyte showed lower bulk resistance values compared to the SPE electrolyte. The ionic conductivities (*σ*) were calculated from Equation (1):(1)σ=L/RA 
where *A* (cm^2^) is the area, *L* (cm) is the thickness of the SPE membrane, and *R* (Ω) is the bulk resistance.

Impedance and ionic conductivity values at different temperatures for both electrolytes are summarized in Table 1. The ionic conductivities of the SPE at 10 °C and 60 °C were $9.17\times 10^{-6}$ S/cm and $5.75\times 10^{-4}$ S/cm, respectively, whereas the ionic conductivities of R-SPE at 10 °C and 60 °C were $1.40\times 10^{-5}\;S/\mathrm{cm}$ and $6.56\times 10^{-4}\;S/\mathrm{cm}$, respectively. These ionic conductivity values are in good agreement with the literature, which discusses solid polymer electrolytes based on the same materials [29,30]. These results clearly indicate that the ionic conductivity of the R-SPE electrolyte s better than that of the SPE. This improvement in the ionic conductivity of the electrolyte can be attributed to the decrease in the crystallinity of the polymer by the application of hot press rolling. This reduction in the crystallinity enhanced the amorphous region in the material, which improved the mobility of lithium ions.

The temperature dependence of the ionic conductivity of both electrolytes, SPE and R-SPE, is shown by Arrhenius curves (Figure 8). The plot illustrates a linear behavior and follows the Arrhenius relationship. The results demonstrate that both electrolytes showed similar temperature-dependent ionic conductivity behavior, showing increases in ionic conductivity with the temperature increase. The increases in ionic conductivity with temperature can be attributed to increases in free volume, polymer segmental mobility, and ionic mobility [31].

The activation energies were calculated by the Arrhenius equation:(2)$$\sigma =\sigma_{0}\;\exp\left(\frac{-E_{a}}{RT}\right)$$
where *σ*, $\sigma_{0}$, *T*, $E_{a}$, *R* are the ionic conductivity, pre-exponential factor, temperature, activation energy, and gas constant, respectively.

The activation energies were calculated as 0.29 eV and 0.31 eV for the R-SPE and SPE, respectively, being in good agreement with the previously reported values for solid polymer electrolytes [32]. The low activation energy of the R-SPE is attributed to an increase in the amorphous region by hot press rolling, which eased lithium-ion movement within the electrolytes. The lower activation energy indicates that lithium ions needed less energy to move within the electrolytes [33].

### 3.2. Direct Current (DC) Polarization and Transference Number

The Li-ion transportation number ($t_{Li}^{+}$) is an important parameter when considering the transportation of lithium ions in solid polymer electrolytes. Coin cells with the configurations Li/SPE/Li and Li/R-SPE/Li were prepared, and DC polarization was performed until a steady-state current was reached (Figure 9c). EIS tests were also conducted before and after the DC polarization for Li/R-SPE/Li and Li/SPE/Li cells (Figure 9a and Figure 9b, respectively). Transference numbers ($t_{Li}^{+}$) were calculated by using the Bruce–Vincent–Evans equation [34]:(3)$$t_{Li}^{+}=\frac{I_{S}\;\left[∆V-I_{O}R_{O}\right]}{I_{O}\;\left[∆V-I_{S\;}R_{S}\right]}$$
where ∆*V* is the applied polarization voltage; $I_{O}$ and $R_{O}$ are the initial current and initial interfacial resistance values before polarization, respectively; and $I_{S}$ and $R_{S}$ are the steady-state current and interfacial resistance values after polarization, respectively. The $R_{O}$ and $R_{S}$ values for both the electrolytes were calculated from impedance results before and after DC polarization (Figure 9a,b).

At room temperature, the transference numbers for the SPE and R-SPE were calculated as 0.12 and 0.18, respectively, which are close to the literature values [21]. The $t_{Li}^{+}$ for the R-SPE was higher than the transference number of the SPE. The remarkable improvement in the lithium-ion transference number can be ascribed to an increase in the grain boundaries. By applying the hot press rolling, the grain boundaries were reformed, which created more pathways for lithium ions to move along. These results are in good agreement with the ionic conductivity results as the movement of lithium ions played an important role in the conductivity improvement.

### 3.3. Linear Sweep Voltammetry (LSV)

Linear sweep voltammetry (LSV) at 60 °C for both the SPE and R-SPE was performed to evaluate their voltage stability windows (Figure 10). Both electrolytes showed stability up to 4.1 V. Slight differences in the results could be due to differences in the OCV values of the cells; otherwise, no significant effect of hot press rolling on the stability voltages was observed.

### 3.4. Symmetrical Cell Cycling

To further investigate the electrochemical stability of electrolytes with a Li-metal anode, symmetric cells with configurations Li/SPE/Li and Li/R-SPE/Li were fabricated and subsequently implemented for plating/striping tests to measure the stability of the electrolytes against lithium dendrite. A current density of 0.1 mA/cm^2^ was applied for the first 500 cycles, followed by 0.2 mA/cm^2^ for the next 500 cycles, then continuing at 0.5 mA/cm^2^ until the cell began to short circuit. This test was performed at 60 °C. Hot press rolled membranes (R-SPE) showed better stability against lithium dendrite growth compared to non-hot press rolled membranes (SPE). In addition, the Li/R-SPE/Li cell performed more cycles (nearly 100 more) compared to the Li/SPE/Li cell (Figure 11).

### 3.5. Galvano-Static Charge–Discharge Cycling

The cycling profiles of the LFP/SPE/Li and LFP/R-SPE/Li cells were tested under a current density of 0.1 C at 60 °C. For the LFP/R-SPE/Li cell, a first discharge specific capacity of 134.3 mAh/g was achieved, whereas for the LFP/SPE/Li cell it was 118 mAh/g (Figure 12a). The LFP/R-SPE/Li cell retained more than 98.5% coulombic efficiency over the first 50 cycles (Figure 12c). The improved cycling performance of the LFP/R-SPE/Li cell can be attributed to the improved ionic conductivity of the electrolyte and interfacial contact between the electrodes and electrolyte after the application of the hot press rolling treatment.

Figure 12b shows the cycling results of the LFP/R-SPE/Li cell at different C-rates of 0.1 C, 0.5 C, and 1 C. At high C-rates, a decline in the capacity of the cell can be seen. This is due to the fast degradation of the battery at high C-rates [35].

The results evidence that the hot press rolling treatment improved the electrochemical performance of the solid polymer electrolyte.

## 4. Conclusions

The ionic conductivity of an electrolyte is greatly affected by the degree of crystallinity of the polymer matrix. In this work, a simple and effective technique of hot press rolling has been shown to improve the performance of solid polymer electrolytes. Rolling at high temperature and pressure causes grain boundary changes in the polymer structure, resulting in an increase in the amorphous region. An unstructured amorphous region helps the movement of lithium ions within the electrolyte.

A significant improvement in the ionic conductivity of the electrolyte was achieved by the application of the hot press rolling technique. It has been experimentally shown that, at 10 °C, the ionic conductivity of the electrolyte increased from $9.17\times 10^{-6}$ S/cm to $1.40\times 10^{-5}$ S/cm, a 53% increase.

The activation energies for both electrolytes were calculated by plotting an Arrhenius curve. For SPEs, it was 0.31 eV and, after hot press rolling treatment, it decreased to 0.29 eV. A lower activation energy means easier Li-ion movement within the electrolyte. Large grain boundaries are unfavorable for Li-ion transport. By applying hot press rolling, the grain boundaries are deformed, which creates more pathways for lithium ions to move along.

An overall improvement in the electrochemical performance of the solid polymer electrolyte has been shown. The smooth surface of the electrolyte provided better contact between the electrode and electrolyte, which improved the performance of the cell. The LFP/R-SPE/Li batteries showed a first discharge capacity of 134.3 mAh/g at 0.1 C, which is better than the capacity shown by LFP/SPE/Li batteries (118 mAh/g). The homogenous structure of the membranes has obvious advantages in improving the performance of the electrolyte.

All the results show that hot press rolling is an effective technique that is easy to implement and improves the performance of solid polymer electrolytes for solid-state lithium-metal batteries.

## Figures and Tables

**Figure 1 polymers-14-00363-f001:**
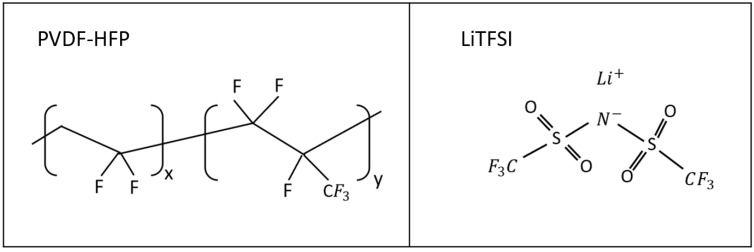
Chemical structures of polymer Polyvinylidene fluoride-co-hexafluoropropylene (PVDF-HFP) and lithium salt Bis(trifluoromethylsulfonyl)amine (LiTFSI) used for the preparation of the solid polymer electrolytes.

**Figure 2 polymers-14-00363-f002:**
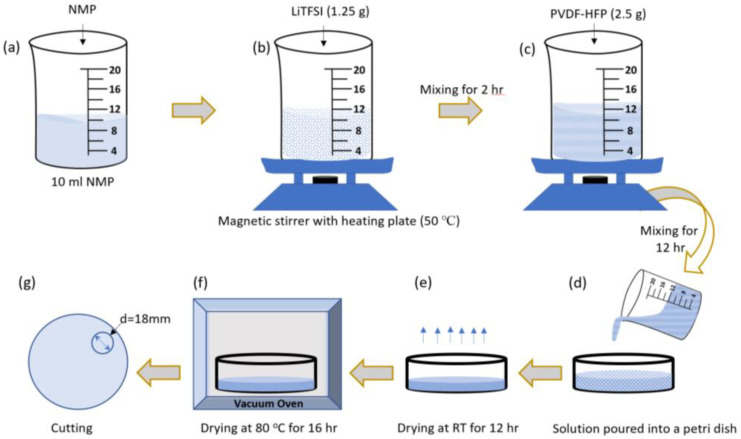
(**a**–**g**) Preparation method for PVDF-HFP/LiTFSI-based solid polymer electrolytes (SPEs).

**Figure 3 polymers-14-00363-f003:**
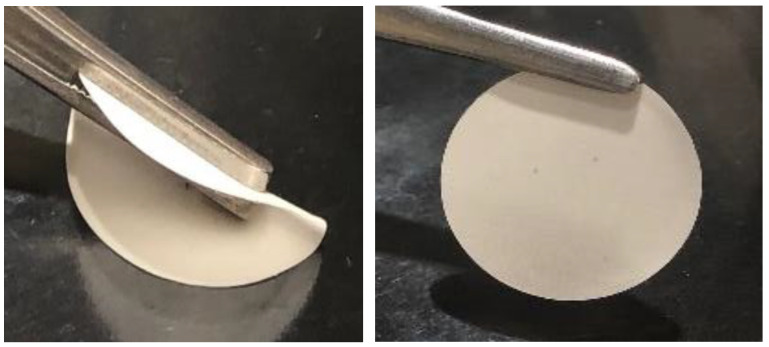
PVDF-HFP/LiTFSI-based solid polymer electrolyte (SPE).

**Figure 4 polymers-14-00363-f004:**
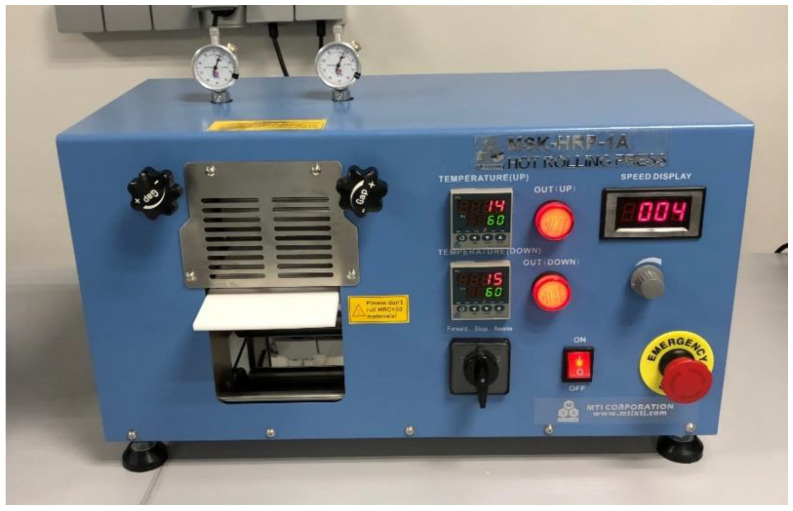
Equipment used for the hot press rolling of solid polymer electrolytes.

**Figure 5 polymers-14-00363-f005:**
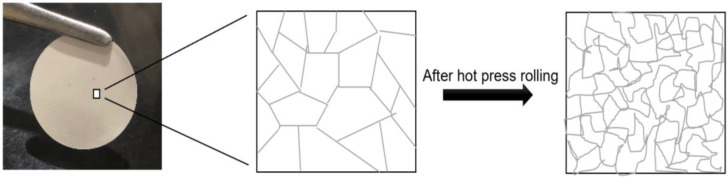
Grain boundary deformation in a solid polymer electrolyte after hot press rolling.

**Figure 6 polymers-14-00363-f006:**
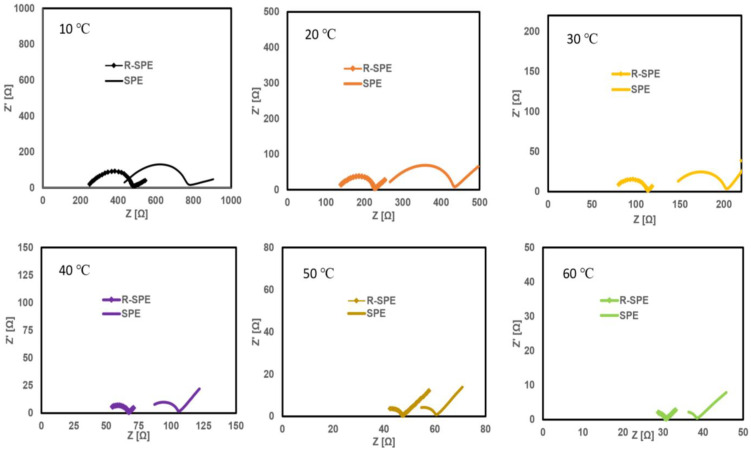
Nyquist plots for SPE and R-SPE electrolytes at different temperatures.

**Figure 7 polymers-14-00363-f007:**
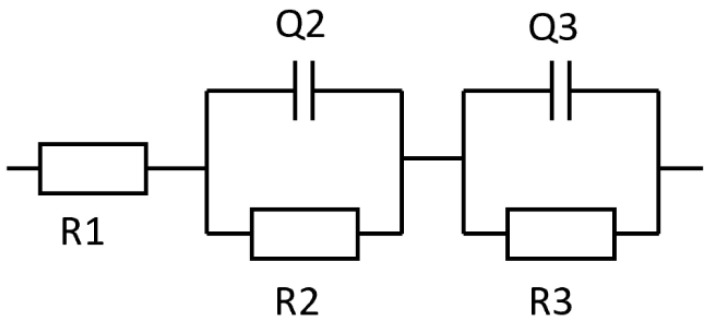
Equivalent circuit proposed to model the ionic conductivity of the solid polymer electrolytes.

**Figure 8 polymers-14-00363-f008:**
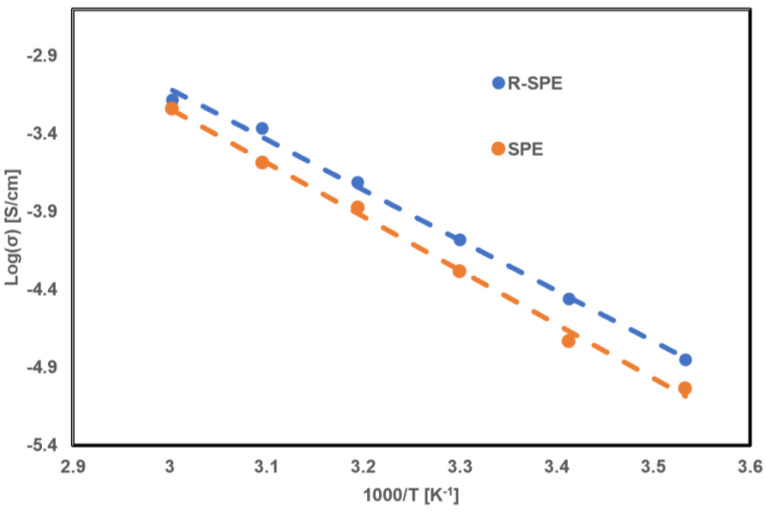
Log σ versus 1000/T plot for SPE and R-SPE electrolytes.

**Figure 9 polymers-14-00363-f009:**
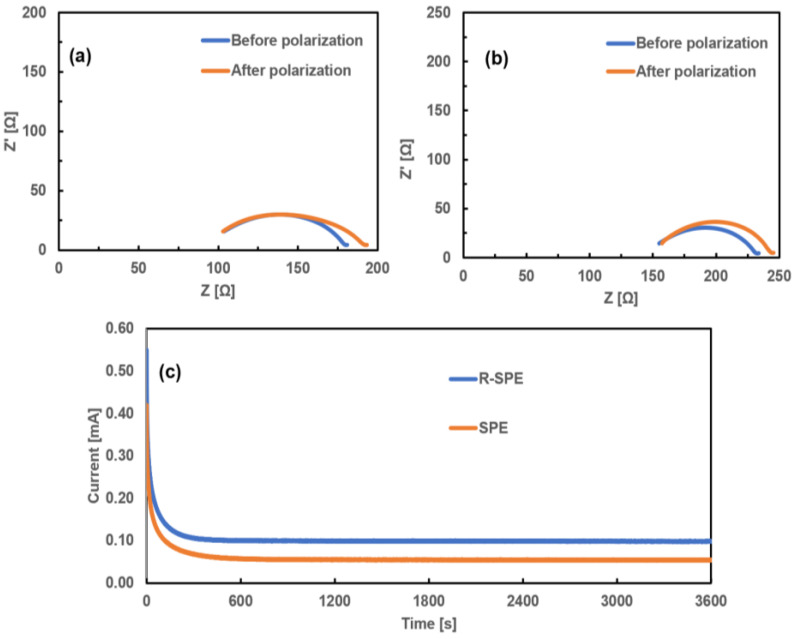
EIS of (**a**) Li/R-SPE/Li and (**b**) Li/SPE/Li cells before and after polarization, and (**c**) DC polarization curve.

**Figure 10 polymers-14-00363-f010:**
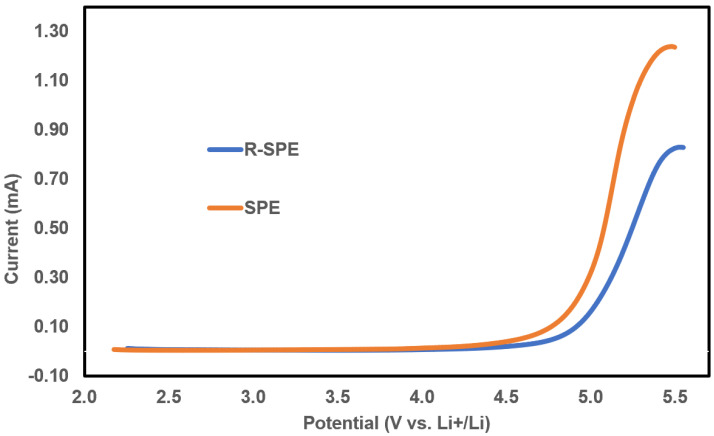
Linear sweep voltammetry (LSV) tests for SS/SPE/Li and SS/R-SPE/Li cells to evaluate their voltage stability windows.

**Figure 11 polymers-14-00363-f011:**
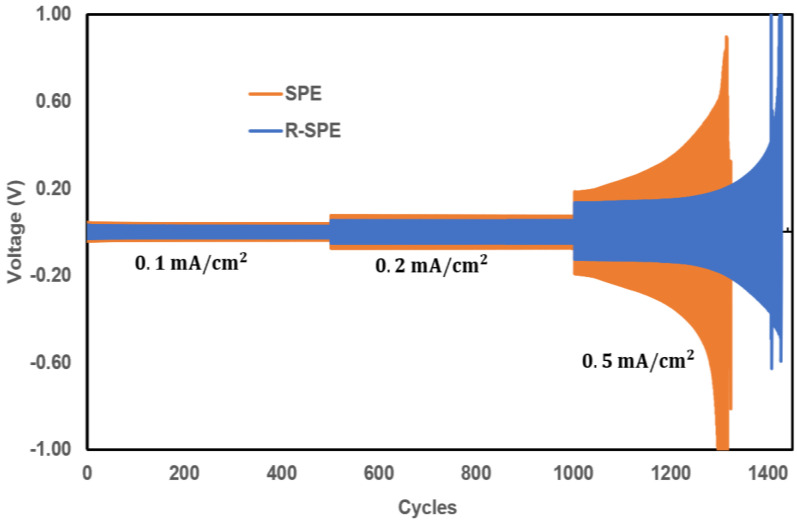
Cycling of symmetrical cells with configurations of Li/SPE/Li and Li/R-SPE/Li.

**Figure 12 polymers-14-00363-f012:**
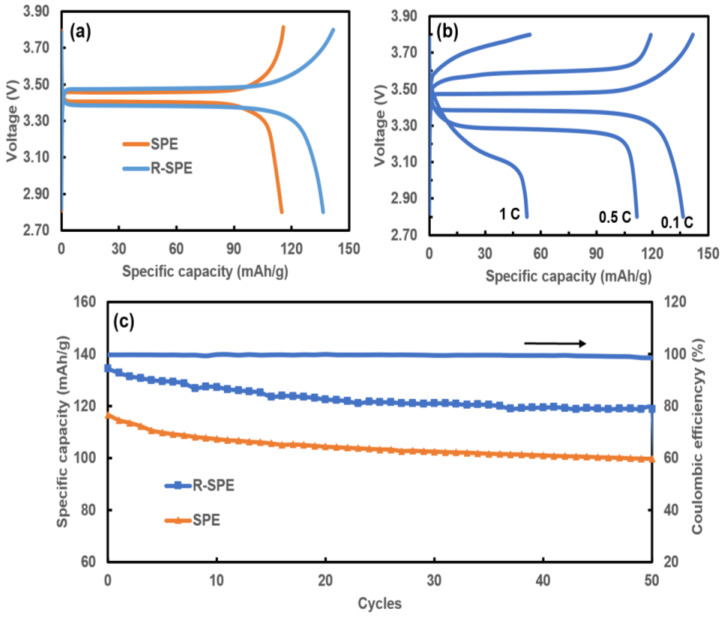
Cycling results for LFP/Li batteries. (**a**) First discharge cycles with SPE and R-SPE electrolytes at 0.1 C and 60 °C, and (**b**) with R-SPE electrolytes at 0.1 C, 0.5 C, and 1 C. (**c**) Long cycle performance with SPE and R-SPE, and coulombic efficiency for a battery with R-SPE.

**Table 1 polymers-14-00363-t001:** Impedance and ionic conductivity values for SPE and R-SPE at different temperatures.

T (°C)	Impedance (Ω) SPE	Impedance (Ω) R-SPE	Ionic Conductivity (S/cm) SPE	Ionic Conductivity (S/cm) R-SPE
10	343	223	9.17 × 10^−6^	1.40 × 10^−5^
20	170	91	1.85 × 10^−5^	3.44 × 10^−5^
30	60	38	5.21 × 10^−5^	8.28 × 10^−5^
40	23	16	1.33 × 10^−4^	1.93 × 10^−4^
50	12	7	2.58 × 10^−4^	4.31 × 10^−4^
60	5	4	5.75 × 10^−4^	6.56 × 10^−4^

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
