# Peer review of "Improved Performance of Solid Polymer Electrolyte for Lithium-Metal Batteries via Hot Press Rolling"

_polymers, 2022, doi:10.3390/polym14030363_

Round 1

Reviewer 1 Report

The manuscript submitted by Poonam Yadav et al investigated the preparation of solid polymer electrolytes comprised of PVDF-HFP and doped with LiTFSI salt. During the preparation of solid polymer electrolytes, hot press rolling was used to enhance the formation of amorphous domains facilitating the transport of Li+. However, the support evidence and characterization are not sufficient and required expansion in the related discussion session. The current form of the manuscript cannot be subjected to the consideration of acceptance before the aforementioned issues are resolved.

In the manuscript, the authors made the claim that more grain boundaries corresponding to amorphous domains were created during the hot press rolling. Albeit, no related structural characterization can be found, the authors may consider adding related characterization to support this assumption.

Ionic conductivity is concentration-dependent, the comparison of conductivity need to involve the influence of salt concentration. In this study, what is the concentration of LiTFSI and how is it when compared with LITFSI in PVDF-HFP but without hot pressing rolling. In the meantime, the SPE with the same compositions has been published (Mathies et al, Solid State Ionic, 2021, 257, 115497), and how are conductivities and cyclability compared with this one.

In the assembly of the battery cell, LFP cathode was used as a cathode which typically yields low energy density and power density, probably not suitable for the next-generation batteries used in EVs with elevated energy density, power delivery, and safety. This combination is not consistent with the motivation mentioned in the Introduction. Did the authors consider assembling batteries with more aggressive electrodes (NCM811 or Ni-rich counterpart) as the demonstration?

Page 9  Figure 9, the axis label overlaps with the figure caption. The authors may consider adjusting the position of the figure.

Reviewer 2 Report

The work «Improved performance of solid polymer electrolyte for lithium-metal batteries via hot press rolling» is actual and interesting. The introduction describes in detail the problem to be solved by the study, possible solutions to this problem, their advantages and disadvantages. The advantages of the proposed approach and selected materials are indicated. In section «Materials and Methods» all stages of the experiment are described in detail, supported by informative Figures in the form of diagrams and photographs. The results obtained are presented in the form of graphs with an explanation of the revealed patterns. The conclusion contains the values of key characteristics and summarizes the research carried out.

I have some comments:

Equation 2 is not an Arrhenius equation.
Description of Figure 9 a and b needs to be strengthened.

 Accept after minor

Round 2

Reviewer 1 Report

The revised manuscript has addressed the comment made by the reviewer in the first review. The questions have been answered with further explanation or additional experiments. The current form of the manuscript is qualified for publication on Polymers after the revisions.